# Mid-Term Mortality in Older Anemic Patients with Type 2 Myocardial Infarction: Does Blood Transfusion sImprove Prognosis?

**DOI:** 10.3390/jcm11092423

**Published:** 2022-04-26

**Authors:** Arthur Hacquin, Alain Putot, Frederic Chague, Patrick Manckoundia, Yves Cottin, Marianne Zeller

**Affiliations:** 1Geriatric Department, Dijon Bourgogne University Hospital, 21000 Dijon, France; patrick.manckoundia@chu-dijon.fr; 2Physiopathologie et Épidémiologie Cérébro-Cardiovasculaires (PEC2), EA 7460, Université de Bourgogne Franche Comté, 21000 Dijon, France; alain.putot@chu-reunion.fr (A.P.); frederic.chague@chu-dijon.fr (F.C.); yves.cottin@chu-dijon.fr (Y.C.); marianne.zeller@chu-dijon.fr (M.Z.); 3Geriatric Department, Réunion University Hospital, 97400 Saint Denis, France; 4Cardiology Department Dijon Bourgogne University Hospital, 21000 Dijon, France; 5INSERM U1093 Cognition Action Plasticité, Université de Bourgogne Franche Comté, 21000 Dijon, France

**Keywords:** type 2 myocardial infarction, anemia, red blood cell transfusion, older patients, mortality

## Abstract

(1) Anemia often predisposes older patients to type 2 myocardial infarction (T2MI). However, the management of this frequent association remains uncertain. We aimed to evaluate the impact of red blood cell transfusion during the acute phase of T2MI in older anemic inpatients. (2) Methods and results: We performed a retrospective study using a French regional database. One hundred and seventy-eight patients aged 65 years or older, presenting with a T2MI and anemia, were selected. Patients were split into two groups: one that received a red blood cell transfusion (≥1 red blood cell unit) and one that did not. A propensity score was built to adjust for potential confounders, and the association between transfusion and 30-day mortality was evaluated with an inverse propensity score weighted Cox model. Transfusion was not associated with 30-day all-cause mortality (propensity score weighted hazard ratio (HR) 1.59 (0.55–4.56), *p* = 0.38). However, 1-year all-cause mortality was significantly higher in the transfusion group (propensity score weighted HR 2.47 (1.22–4.97), *p* = 0.011). (3) Conclusion: Our findings in older adults with anemia suggest that blood transfusion in the acute phase of T2MI could not be associated with improved short-term prognosis. Prospective studies are urgently needed to assess the impact of transfusion on longer-term prognosis.

## 1. Introduction

Myocardial infarction (MI) is a leading cause of hospitalization and mortality in older adults. Different types of MI have been described in the recent literature. While type 1 MI (T1MI) has been widely studied, type 2 myocardial infarction (T2MI), defined as an imbalance between myocardial oxygen supply and demand that is unrelated to acute atherothrombosis, remains only partially explored. There are various underlying pathophysiological mechanisms in T2MI, including sustained tachyarrhythmia, severe hypertension, septic events, or severe anemia [1]. Chronic anemia, which is a proven major predisposing factor for T2MI, is a frequent issue in the older population, in which frailty and acute or chronic bleeding are more common [1,2,3,4].

Although the prognosis for T2MI is worse than for T1MI, the optimal treatment and management strategies remain unknown. The 4th Universal Definition of MI recommends treating the underlying origin of ischemic imbalance of oxygen supply and demand [1]. In anemic patients, the treatment options include red blood cell transfusions. Transfusion is often indicated in anemia when the hemoglobin level is below 10 g/dL, but there are wide variations according to clinical practice [5]. In patients with coronary artery disease, the current recommendations are to transfuse blood when the hemoglobin level is below 8 g/dL [6]. However, these guidelines are not specifically designed for T2MI. The recent REALITY trial in patients with acute MI and anemia proved that a restrictive transfusion strategy (hemoglobin ≤ 8 g/dL) yielded a noninferior rate of cardiovascular events after 30 days, when compared with a liberal transfusion strategy (hemoglobin ≤ 10 g/dL) [7]. Furthermore, it has recently been suggested that blood transfusions might be harmful for older patients when administered during the acute phase of MI [8]. However, there is currently no specific data about the prognostic impact of transfusion at the acute phase of a T2MI with anemia during the hospitalization stay. To address this frequent but poorly understood issue, we assessed the prognosis of red blood cell transfusion in older anemic patients, using data from the RICO survey, which is a large database of acute MI cases.

## 2. Materials and Methods

### 2.1. Study Design

We performed an analytic retrospective study using data collected from the French regional *obseRvatoire des Infarctus de Côte-d’Or* (RICO), which is an ongoing prospective survey that collects data relative to patients hospitalized for acute MI in the intensive care units (ICUs) of an eastern region in France. Population characteristics have been previously described [9]. Adjudication for T1MI and T2MI, based on the 3rd Universal Definition of MI, was systematically performed, as previously described [3,10]. Patients with non-ischemic myocardial injury, Tako Tsubo syndrome, and type 3, type 4, and type 5 myocardial infarction are not included in the registry.

### 2.2. Outcomes

To assess the prognostic impact of transfusion, several outcomes were considered, including all-cause and cardiovascular (CV) mortality, at 30 days and 1 year. The duration of stay in the ICU, and the rates of re-hospitalization for heart failure or recurrent MI at 1-year follow-up, were also recorded as explanatory variables.

### 2.3. Patients

Patients were included if they had been admitted to the CCU between 1 January 2005 and 31 December 2017, were aged 65 years or more, had undergone coronary angiography, and had presented with anemia during their hospitalization for T2MI. Anemia was defined as a nadir hemoglobin level below 10 g/dL during the index hospital stay. Chronic anemia, as a cause of T2MI, was considered as a co-morbid condition, but not as an inclusion criterion. Patients with T1MI, i.e., atherothrombotic events at coronary angiography, and patients lost to 1-year follow-up were excluded (*n* = 36). Other exclusion criteria included coronary artery bypass graft or any other surgery during hospitalization, and palliative care before blood transfusion. We compared patients with at least one red blood cell transfusion to those with no red blood cell transfusion during the hospital stay.

Patients received acute medications in the setting of routine clinical practice and in agreement with current guidelines.

### 2.4. Data Collection

Demographic data, CV risk factors, medical history, chronic treatments, and heart failure assessed using Killip class were collected on admission [11]. Obesity was defined as a BMI > 30 kg/m^2^. Left ventricular ejection fraction (LVEF) was assessed via echocardiography on admission. The LVEF cutoff was set at 40% for more clinical relevance. The in-hospital transfusion of ≥1 units of packed red blood cells was recorded.

Vascular history included myocardial infarction, peripheral arterial disease, or stroke. A history of thromboembolic events included pulmonary embolism or deep vein thrombosis. Aortic stenosis was diagnosed according to current guidelines (i.e., peak aortic velocity, mean pressure gradient across the aortic valve, and aortic valve area) using continuous-wave Doppler ultrasound [12].

### 2.5. Biological Data

Blood samples were taken on admission to measure hemoglobin and C-reactive protein (CRP) levels, plasma *n*-terminal pro-brain natriuretic peptide (NT pro-BNP), and serum creatinine. Hemoglobin was also measured every day during the hospital stay. The nadir hemoglobin level and a drop in hemoglobin level were defined as the lowest hemoglobin value during hospitalization and a drop of at least 3 g/dL since admission, respectively. The estimated glomerular filtration rate was calculated using the Chronic Kidney Disease Epidemiology Collaboration formula (CKD-EPI). Cardiac troponin I peak was assessed by sampling every 8 h in the first 48 h after admission.

The present study is in accordance with the declaration of Helsinki and has been approved by the ethics committee of Dijon Bourgogne University Hospital. Each patient provided written consent before participation.

### 2.6. Statistical Analysis

#### 2.6.1. Missing Values

Variables missing more than 20% of values were excluded from the analyses. There were no missing values for primary and secondary outcomes, except for duration of ICU stay. Given the large number of patients in the initial cohort, we considered that they were missing at random, and imputed 10 datasets using a multiple imputation with the predictive mean matching method [13,14].

#### 2.6.2. Description of Covariates

Continuous variables were reported as medians (interquartile range (IQR)), and categorical variables as numbers (percentage). The Student’s *t*-test or Wilcoxon rank test were used to compare continuous data, $\chi^{2}$ or Fisher exact tests were used for categorical data, as appropriate, and the log-rank test was used for survival data.

#### 2.6.3. Propensity Score

Transfusion can be related to certain confounding factors, which could potentially lead to differences between the two groups. Given the non-randomized design of this study, we constructed a propensity score for each patient and applied a stabilized inverse probability weighting (SIPW) within each dataset to identify and control for confounding factors [15]. Given the small sample size, the SIPW method was used for matching on the propensity score to avoid a loss of patient data during comparison, and the stabilization procedure aimed to avoid an extreme value bias [15,16]. We first performed a bivariate analysis to test the differences between the transfusion and no transfusion group. All variables with a *p*-value < 0.05 and/or standard mean deviation > 0.1 (Table 1) were included to build propensity scores using non-parsimonious logistic regression, where transfusion was the explanatory variable. A lack of or negligible collinearity between covariates was checked in all propensity score models. Then, we selected the best score to predict the transfusion, using receiver operating characteristic (ROC) curves (Appendix A) and love plots (Appendix A). The ability of the propensity score to achieve group similarity was verified by comparing the standard mean deviation of each variable, prior to and after adjustment on the score (Appendix A).

Survival analyses were performed using SIPW-adjusted Kaplan–Meier curves. [17] Cox proportional hazard regression was then weighted on SIPW propensity score to construct a Marginal Structural Cox model to evaluate the association of transfusion on short- and mid-term mortality. Proportional risk assumption was graphically checked using weighted residuals. Statistical tests were two-tailed, and *p* < 0.05 was considered significant. Data management and statistical analyses were performed with RStudio version 1.3.1073 (RStudio Team (2020). RStudio: Integrated Development Environment for R. RStudio, PBC, Boston, MA, USA).

## 3. Results

### 3.1. Population

A total of 2135 patients were identified in the database as having MI; 1640 were adjudicated as T1MI, and 20 were adjudicated as type 3 or 4 or 5 MI. Therefore, 475 patients were classified as T2MI. Of the 201 patients who had nadir hemoglobin < 10 g/dL, 23 underwent a coronary artery bypass graft or a surgical intervention. Finally, 178 patients were analyzed; 86 received a red blood cell transfusion and 92 did not (Figure 1).

Characteristics of the population pre-weighting on the propensity score are shown in Table 1.

Median age, female sex, and CV risk factor rates, including obesity and chronic treatments, were similar for the two groups. However, the aortic stenosis rate was twice as high in the transfusion group (30 vs. 15%, *p* = 0.027).

In patients with no transfusion, systolic and diastolic blood pressure on admission were higher (140 vs. 123 mmHg, *p* = 0.005, and 72 vs. 64 mmHg, *p* < 0.001, respectively), and LVEF (50 vs. 45%, *p* = 0.04) was lower than in patients with transfusion. Heart failure on admission was similar for the two groups.

As expected, transfused patients had lower hemoglobin levels on admission and nadir hemoglobin levels than non-transfused patients (9.9 vs. 10.7 g/dL, *p* = 0.001, and 7.8 vs. 9.3 g/dL, *p* < 0.001, respectively). In addition, significantly more transfused patients had a drop in hemoglobin (89.5 vs. 45.7%, *p* < 0.001).

Peak troponin Ic was almost three times higher in the transfused group (9.8 vs. 3.1 ng/mL, *p* = 0.008). No difference between the two groups was spotted for CRP values or biomarkers of renal function. Coronary angiography was less frequently used in patients who had received a blood transfusion. (82.6 vs. 94.6%, *p* = 0.022).

ROC curve analysis found an Area Under the Curve of 0.89 [0.84–0.94] (Appendix A). The initial imbalance between the two groups before weighting disappeared after propensity score adjustment (Appendix A).

### 3.2. Outcomes

ICU duration was similar for the two groups (*p* = 0.69), as were the rates of all-cause and CV death at 30 days (*p* = 0.37 and 0.89, respectively) (Table 2).

After propensity score weighting, transfusion was not associated with 30-day mortality (propensity score weighted HR (95% CI): 1.59 (0.55–4.56), *p* = 0.38) (Table 3).

At 1-year follow-up, 60 patients had died; more deaths had occurred in the transfusion group than in the no transfusion group (*p* = 0.02) (Table 2). Transfusion remained associated with 1-year mortality after weighting (propensity score weighted HR (95% CI): 2.47 (1.22–4.97), *p* = 0.011) (Table 3). The 1-year survival-adjusted Kaplan–Meier curve is reported in Figure 2.

Given the specific association of transfusion with outcomes in the oldest patients with acute MI, [8] we further performed a stratified analysis based on age (threshold at 80 years) (Table 3). However, the relationship between transfusion and the outcome was not significant, regardless of the age group.

At the 1-year follow-up, there was only a trend toward higher CV mortality in the transfusion group (*p* = 0.08).

Re-hospitalization for CV causes, including recurrent MI or heart failure, were not significantly higher in the transfusion group (*p* = 1.0 and *p* = 0.08, respectively).

Given the higher rate of aortic stenosis in transfused patients, we tested the introduction of this variable into our models. Even after adjustment for aortic stenosis, there was still a trend toward higher all-cause mortality associated with transfusion (adjusted HR (95% CI): 1.97 (0.94–4.14), *p* = 0.07), and no effect on CV mortality (adjusted HR (95% CI): 1.13 (0.23–5.48), *p* = 0.9).

## 4. Discussion

Anemia is a common condition in older patients hospitalized with acute MI and is associated with poor outcomes [18]. Given that T2MI is caused by a mismatch between oxygen supply and demand, red blood cell transfusion, through increased oxygen delivery, should in theory be a particularly attractive therapeutic option. However, our findings, obtained from a retrospective population-based study and by using multivariate analysis to minimize the effect of confounding factors, suggest that there are potentially harmful effects of transfusion in this type of MI.

The risk–benefit balance of transfusions remains unclear, particularly in elderly patients with acute MI [19], and oxygen delivery may not be increased with transfusion. Transfusion may actually increase platelet activation and aggregation and produce vasoconstriction, thus inducing a prothrombotic state and recurrent ischemic events [20]. Moreover, transfusion has potential risks, including infection, alloimmunization, circulatory overload, and acute lung injury [20].

Multivariate analysis showed that no significant association was found between transfusion and short-term mortality. This was confirmed via visual analysis of the early points of the Kaplan–Meier survival curves, which contrasted with a significantly increased rate of mortality at the 1-year follow-up. This apparent discrepancy may be explained by the increase in the number of deaths at long-term follow-up improving the achievement of significance. Comparatively, there were reduced cases of in-hospital death rate, which showed only a trend in mortality toward a higher transfused group (30-day mortality 7 (7.6) vs. 11 (12.8), respectively, in the no transfusion and transfusion groups, *p* = 0.37; 1-year mortality 23 (25.0) vs. 37 (43.0), *p* = 0.02). In addition, in older patients, multiple co-morbidities may impact on long-term follow-up, including chronic renal failure (16 (17.6) vs. 19 (22.9), respectively, in the no transfusion and transfusion groups, *p* = 0.494), and aortic stenosis (14 (15.2) vs. 26 (30.2) *p* = 0.027).

Randomized trials investigating blood transfusion in acute MI are scarce. The recent REALITY interventional study included 668 patients with acute MI and hemoglobin levels between 7 and 10 g/dL. A liberal blood transfusion strategy (triggered by hemoglobin ≤ 10 g/dL) showed a trend toward increased all-cause and CV deaths when compared to a restrictive strategy (triggered by hemoglobin ≤ 8 g/dL) [7]. It has also been suggested that transfusion effects could depend on patient age; older patients, who are more prone to diffuse coronary artery lesions, could be more sensitive to an anemia-related supply and demand mismatch [8].

A recent meta-analysis [21] included 6630 patients on six articles, of whom five had acute MI. They found that restrictive transfusion tended to have a high risk of in-hospital mortality. However, the mean age (69.0 to 79.5 y) was much younger than in the present study. Moreover, although the type of MI was not reported, it is probable that most were T1 MI, given the inclusion period (2001 to 2021). These differences, in addition to the type of study (i.e., observational vs. randomized), may at least partly explain the difference in hospital prognosis.

A systematic review showed that the effects of transfusion were beneficial, or at least neutral, when compared to non-transfusion when the hemoglobin level was <8 g/dL, but that they were harmful when hemoglobin was >11 g/dL [22,23]. Another study showed no difference in mortality in restrictive and liberal transfusion groups [24]. However, the included studies included a majority of T1MI patients and did not specifically focus on T2MI.

Compared with T1MI, T2MI is generally associated with higher all-cause mortality (twice as high in some studies [19]) and higher rates of major CV events [25]. Interestingly, consistent data suggest that the higher rate of death would be mainly due to non-CV death from a yet-to-be-identified origin [26].

To the best of our knowledge, our study is the first to focus on the prognosis of transfused patients with anemia during hospitalization for T2MI. While blood transfusion was not associated with 30-day mortality in our results, 1-year all-cause mortality was higher in the transfused group in the unadjusted analysis, as well as in the SIPW-model analysis. Although there were some differences between the groups at baseline, the use of propensity-score-based adjustments helped to adequately reduce most of them after weighting, thus providing relative confidence in the findings.

Comorbidities did not differ between groups, except for aortic stenosis, which was much more frequent in the transfusion group. Patients with severe aortic stenosis are particularly prone to anemia because they are frequently on antiplatelet and/or anticoagulant treatment, which increases the risk of bleeding [27,28]. Moreover, severe aortic stenosis is associated with a worse prognosis, even for a mild degree of anemia. This could partly explain the increased risk, as suggested by the decrease in HR values after adjustment on aortic stenosis. LVEF, which is a major prognostic factor for death after T1MI, was higher in the transfusion group, and thus it may not contribute to the worse prognosis [29,30].

There were some limits to this study. Most of the confounding factors were controlled by propensity matching–inverse probability weighting, and the initial imbalance between the two groups before weighting was considerably reduced after PS adjustment, as shown in the love plot (Appendix A) and in the Appendix A, and most SMDs were <0.01. However, some variables, such as nadir hemoglobin and diastolic blood pressure, did not remain strictly balanced, and we assume that this may limit the reliability of the findings.

The retrospective nature of the study and the small sample size limit the conclusions that can be drawn from our findings. However, the data are in agreement with most studies, including a recent meta-analysis [21]. Moreover, some missing values may have limited the validity of the analysis. However, we did perform multiple imputations for all of the covariates, and then calculated, in agreement with the reference method, a propensity score in each data set, to limit the impact of missing data [14]. Moreover, the small sample size may have reduced the power of the models. However, we assessed the validity of the propensity score, using a ROC curve (Appendix A) and a love plot (Appendix A), comparing the standard mean deviation prior and after adjustment on the score (Appendix A).

We were not able to gather precise information on the acute management treatment of myocardial infarction; however, patients received medication within the setting of a routine clinical practice, and in agreement with current guidelines.

Although the medical files of each deceased patient in this older co-morbid population were manually reviewed, the precise cause of death was not available for most patients, and therefore it was not reported in this article.

Finally, adjudication between T2MI and T1MI is complex, and a limited number of patients without coronary angiography may have had a thromboembolic event. However, the systematic adjudication procedure, as previously described [3], strongly suggests the reliability of our results.

## 5. Conclusions

Our retrospective study on a large MI database suggests that red blood cell transfusion in older adults with anemia during hospitalization for T2MI has no significant impact on short-term mortality, but it could have a deleterious effect on their longer-term prognosis. More exploration is warranted, such as sub-group analysis to define the prognostic value in sub-groups, and optimal treatment strategies in older T2MI patients with anemia.

## Figures and Tables

**Figure 1 jcm-11-02423-f001:**
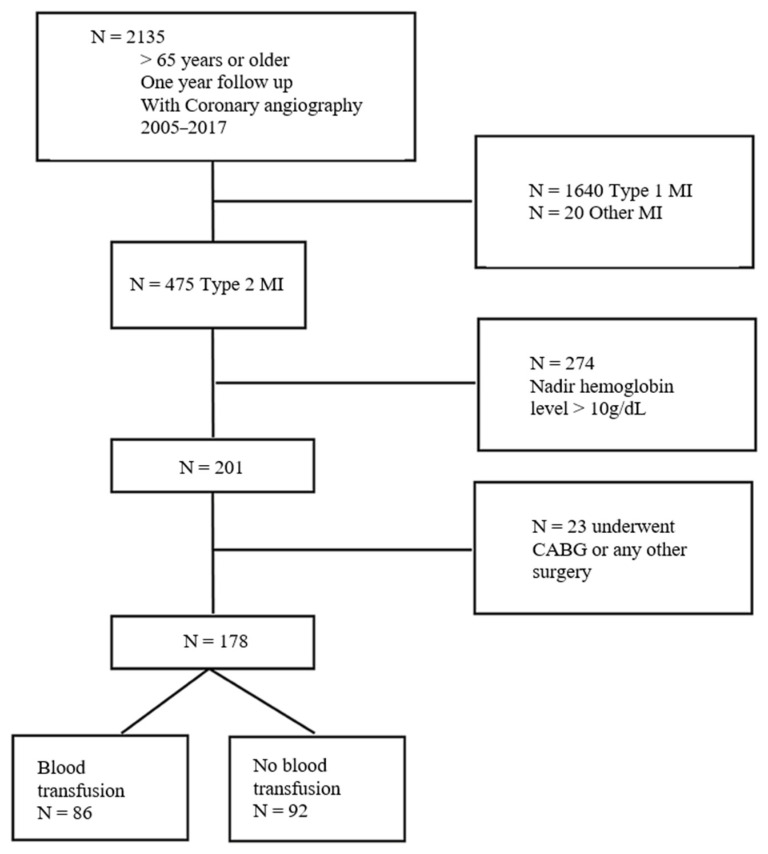
Flow chart of the study population. MI, myocardial infarction; CABG, coronary artery bypass graft.

**Figure 2 jcm-11-02423-f002:**
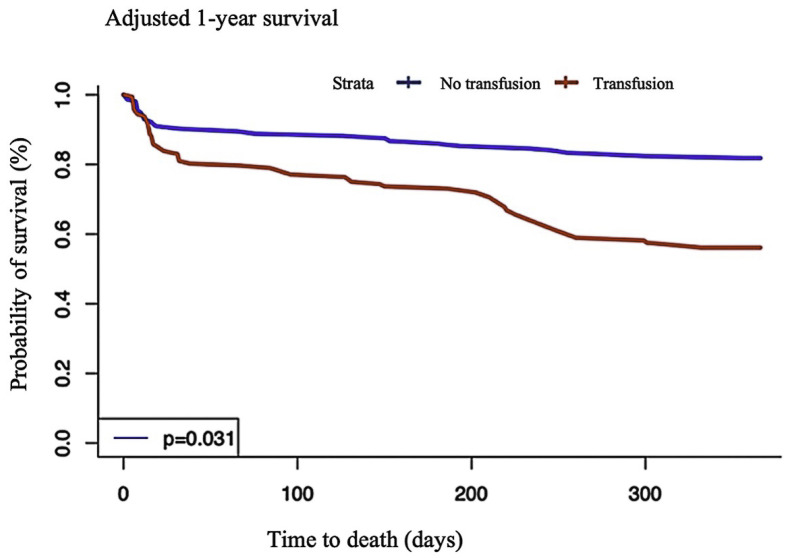
One-year survival Kaplan–Meier curves adjusted on the propensity score.

**Table 1 jcm-11-02423-t001:** Baseline characteristics (*n* (%) or median [interquartile range]). BMI: body mass index; CAD: coronary artery disease; SBP: systolic blood pressure; DBP: diastolic blood pressure; CV: cardiovascular; LVEF: left ventricular ejection fraction; CKD-EPI: Chronic Kidney Disease Epidemiology Collaboration; NT-proBNP: *n*-terminal pro-brain natriuretic peptide.

	No Transfusion (*n* = 92)	Transfusion (*n* = 86)	*p* Value
**Demographic data**			
Age (year)	81.5 (76,75–86)	83 (79–86)	0.22
Age > 80 (year)	51 (55.4)	60 (69.8)	0.07
Female	52 (56.5)	48 (55.8)	1.00
BMI (kg/m^2^) (*n* = 176)	25 (22–29)	24 (23–27)	0.15
Obesity (*n* = 176)	18 (19.6)	9 (10.5)	0.14
**CV risk factors**			
Hypertension (*n* = 178)	73 (79.3)	72 (83.7)	0.58
Diabetes (*n* = 178)	30 (32.6)	27 (31.4)	0.99
Dyslipidemia (*n* = 177)	52 (56.5)	47 (55.3)	0.99
Family history of CAD (*n* = 167)	13 (14.9)	16 (20.0)	0.511
Smoking (*n* = 168)	7 (7.6)	5 (5.8)	0.859
**Medical history**			
Vascular history (*n* = 178)	22 (23.9)	27 (31.4)	0.343
Myocardial infarction (*n* = 178)	22 (23.9)	16 (18.6)	0.496
Coronary artery bypass graft (*n* = 178)	9 (9.8)	8 (9.3)	1.000
Kidney disease (*n* = 174)	16 (17.6)	19 (22.9)	0.494
Thrombo-embolic event (*n* = 176)	12 (13.2)	7(8.2)	0.415
Atrial fibrillation (*n* = 166)	23 (26.7)	21 (26.2)	1.000
Aortic stenosis (*n* = 178)	14 (15.2)	26 (30.2)	0.027
Neurocognitive disorder (*n* = 172)	5 (5.6)	6 (7.3)	0.873
Neoplasia (*n* = 173)	27 (30.0)	24 (28.9)	1.000
**Chronic treatments**			
Aspirin (*n* = 178)	32 (34.8)	34 (39.5)	0.617
Other antiplatelet (*n* = 178)	15 (16.3)	19 (22.1)	0.429
Vitamin K inhibitor (*n* = 178)	28 (30.4)	17 (19.8)	0.143
Oral anticoagulant (*n* = 178)	0 (0.0)	1 (1.2)	0.973
Calcium inhibitor (*n* = 178)	24 (26.1)	26 (30.2)	0.654
Angiotensin Receptor Blocker (*n* = 178)	32 (34.8)	21 (24.4)	0.178
Angiotensin Converting Enzyme inhibitor (*n* = 178)	19 (20.7)	23 (26.7)	0.435
**Clinical data on admission**			
Heart rate (b/min) (*n* = 162)	86 (72–100)	82 (70–100)	0.504
SBP (mmHg) (*n* = 163)	140 (119–156)	123 (120–144)	0.005
DBP (mmHg) (*n* = 163)	72 (63–81.5)	64 (55–74)	<0.001
Heart failure (*n* = 178)	51 (55.4)	44 (51.8)	0.73
LVEF (%) (*n* = 177)	45 (35–60)	50 (40–60)	0.039
LVEF > 40% (*n* = 177)	63 (68.5)	69 (80.2)	0.105
**Biological data**			
Hemoglobin at admission (g/dL) (*n* = 178)	10.75 (9.9–12)	9.9 (8.7–11.4)	0.001
Nadir hemoglobin level (g/dL) (*n* = 178)	9.3 (8.9–9.7)	7.8 (7.3–8.3)	<0.001
Drop in hemoglobin (*n* = 178)	42 (45.7)	77 (89.5)	<0.001
Creatinine (µmol/L) (*n* = 176)	102 (72–139)	112 (81–147)	0.410
e-GFR (CKD-EPI) < 60 mL/min/1.73 m^2^ (*n* = 176)	56 (60.9)	55 (64.0)	0.787
C reactive protein > 3 mg/L (*n* = 175)	74 (82.2)	73 (85.9)	0.650
NT-proBNP (pg/mL) (*n* = 169)	6632 (2018–15,993)	6068 (3131–13,824)	0.709
Troponin Ic peak (ng/mL) (*n* = 175)	3.1 (0.96–9.77)	9.8 (2.8–22)	0.008
Coronary angiography (*n* = 178)	87 (94.6)	71 (82.6)	0.022

**Table 2 jcm-11-02423-t002:** Outcomes *n*(%) or median [IQR]. ICU: intensive care unit; CV: cardiovascular; MI: myocardial infarction.

	No Transfusion	Transfusion	
ICU stay duration (d)	4.0 (3.0–7.0)	5.0 (3.0–6.5)	*p* = 0.69
**30-day**			
All-cause death	7 (7.6)	11 (12.8)	*p* = 0.37
CV death	7 (7.6)	8 (9.3)	*p* = 0.89
**1-year**			
All-cause death	23 (25.0)	37 (43.0)	*p* = 0.02
CV death	13 (14.1)	22 (25.6)	*p* = 0.08
Recurrent MI	3 (3.3)	3 (3.5)	*p* = 1.0
Re-hospitalization for heart failure	67 (72.8)	51 (59.3)	*p* = 0.08

**Table 3 jcm-11-02423-t003:** Cox proportional hazard regression analysis of red blood cell transfusion for mortality. HR: hazard ratio; CI: confidence interval; SIPW: stabilized inverse probability weighting.

	30-Day Mortality		1-Year Mortality	
	HR [95% CI]	*p*	HR [95% CI]	*p*
Unadjusted	1.39 (0.61–3.18)	0.42	1.89 (1.12–3.19)	0.02
SIPW-adjusted				
All patients	1.59 (0.55–4.56)	0.38	2.47 (1.22–4.97)	0.01
Stratified on age				
≤80 y	1.70 (0.37–7.81)	0.49	2.30 (0.74–7.23)	0.14
>80 y	1.55 (0.38–6.33)	0.53	1.94 (0.76–4.98)	0.16

## Data Availability

Restrictions apply to the availability of these data. Data was obtained from the French regional *obseRvatoire des Infarctus de Côte-d’Or* (RICO) and are not available for sharing.

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
