# Peer review of "Mid-Term Mortality in Older Anemic Patients with Type 2 Myocardial Infarction: Does Blood Transfusion sImprove Prognosis?"

_jcm, 2022, doi:10.3390/jcm11092423_

Round 1

Reviewer 1 Report

The authors conducted a retrospective study to assess whether blood transfusion improve prognosis in older anemic patients with type 2 myocardial infarction (T2MI). The result showed that blood transfusion in the acute phase of T2MI was not associated with improved short-prognosis and could be associated with a deleterious impact on longer-term prognosis in older adults with anemia. However, there existed many weaknesses in this study.

  1. This was a retrospective study with small sample size and the presence of some missing values, the conclusion that blood transfusion was not associated with improved short-prognosis and could be associated with a deleterious impact on longer-term prognosis in older anemic patients with T2MI should be questioned.
  2. How did the author define anemia-related T2MI and anemia comorbidity T2MI? The causes of T2MI include coronary spasm, coronary embolism, anemia, arrhythmia, hypertension, or hypotension, et al. The term “anemia-related myocardial infarction” may not be appropriate for this study.
  3. Did patients in both groups receive standard treatment for myocardial infarction? This should be described in the study as it may have an effect on the outcome.
  4. Although the confounding factors were controlled by the method of propensity matching-inverse probability weighting, there were still some important variables that were imbalanced, which would affect the reliability of the results.
  5. It is suggested that the reasons for transfusion associated with 1-year mortality but not 30-day mortality should be explained in the discussion section.

A recent meta-analysis demonstrated that restrictive transfusion tended to have a higher risk of in-hospital mortality in patients with acute myocardial infarction and anemia (PMID: 34869640). However, the authors found that blood transfusion in the acute phase of T2MI was not associated with poor short-prognosis. Thus, comparisons with other studies should be discussed more detail in the discussion section.   

Reviewer 2 Report

Hacquin et al. performed an interesting analysis concerning the role of transfusion in type 2 MI. I have the following concerns:

1. Are the patients lost to follow-up at 1 year included in 30 day mortality analysis. Please specify.

2. Could the authors explain why they chose 80 as a threshold? In addition, the authors should clarify number of patients older than 80 in tables concerning analysis using this disambiguation.

3. Perhaps the biggest issue in such analysis is appropriate adjustment for confounders. First and perhaps the biggest are nadir hemoglobin levels alongside hemoglobin levels at admission. Namely, they are significantly lower in the transfusion group (which is expected), thus confounding us whether the differences are because of bigger blood loss or because of transfusion itself. The authors should perform analysis in which they will compare patients with the same amount of hemoglobin. On the other hand, after adjustment for aortic stenosis, mortality was not significantly different among groups, thus questioning the conclusion. In this sense, the authors should adjust the sentence "trend toward ..." with more appropriate nomenclature for such observation (https://www.ncbi.nlm.nih.gov/pmc/articles/PMC6440716/). 

Round 2

Reviewer 1 Report

The manuscript was improved, and i recommended to accept it.

Reviewer 2 Report

No further comments.